# A quantitative description of light-limited cyanobacterial growth using flux balance analysis

**Rune Höper**[1], **Daria Komkova**[1], **Tomáš Zavřel**[2], **Ralf Steuer**[1,3]*

**1** Institute for Biology, Theoretical Biology (ITB), Humboldt-University of Berlin, Berlin, Germany,
**2** Department of Adaptive Biotechnologies, Global Change Research Institute of the Czech Academy of Sciences, Brno, Czechia, **3** Peter Debye Institute for Soft Matter Physics, Universität Leipzig, Leipzig, Germany

* ralf.steuer@uni-leipzig.de

**Data Availability Statement:** The manuscript and its Supporting information contains all raw data required to replicate the results of our study.Scripts and further instructions are available as S4 File and on GitHub at https://github.com/krotlkpt/syn-

## Abstract

The metabolism of phototrophic cyanobacteria is an integral part of global biogeochemical cycles, and the capability of cyanobacteria to assimilate atmospheric $CO_2$ into organic carbon has manifold potential applications for a sustainable biotechnology. To elucidate the properties of cyanobacterial metabolism and growth, computational reconstructions of genome-scale metabolic networks play an increasingly important role. Here, we present an updated reconstruction of the metabolic network of the cyanobacterium *Synechocystis* sp. PCC 6803 and its quantitative evaluation using flux balance analysis (FBA). To overcome limitations of conventional FBA, and to allow for the integration of experimental analyses, we develop a novel approach to describe light absorption and light utilization within the framework of FBA. Our approach incorporates photoinhibition and a variable quantum yield into the constraint-based description of light-limited phototrophic growth. We show that the resulting model is capable of predicting quantitative properties of cyanobacterial growth, including photosynthetic oxygen evolution and the ATP/NADPH ratio required for growth and cellular maintenance. Our approach retains the computational and conceptual simplicity of FBA and is readily applicable to other phototrophic microorganisms.

## Author summary

Phototrophic microorganisms, including cyanobacteria, have significant potential as a resource for a circular economy. Computational strain design and other applications increasingly make use of genome-scale metabolic reconstructions. Genome-scale metabolic reconstructions, together with computational methods of linear programming and constraint-based analysis, allow us to probe the metabolic capabilities and the maximal growth rate of the respective organisms. While such computational methods are well established for heterotrophic bacteria, the description of phototrophic metabolism gives rise to additional challenges due to the use of light as the primary source of energy. In this work, we present a novel computational approach to incorporate light absorption and

growth-fit and at https://github.com/krotlkpt/syn6803.

**Funding:** This work was funded by the DFG grant "Eating the Sun: the path from single cell growth to productive ecosystems" (Grant 453048493 to RS) and by the Ministry of Education, Youth and Sports of CR (Grant LUAUS24131 to TZ). The funders had no role in study design, data collection and analysis, decision to publish, or preparation of the manuscript.

**Competing interests:** The authors have declared that no competing interests exist.

light utilization into a constraint-based description of phototrophic growth. Our approach allows us to describe the variable quantum yield, the maximal photosynthetic capacity, and the relative contribution of linear and cyclic electron transport, and is supported by quantitative data from cyanobacterial growth experiments. We argue that our approach advances the quantitative analysis of light-limited phototrophic growth and can be readily applied to other phototrophic microorganisms.

## Introduction

Oxygenic photosynthesis is one of the most important biological processes on our planet and drives primary production in almost all ecosystems. To this day, cyanobacteria, the evolutionary inventors of oxygenic photosynthesis, remain an integral part of global biogeochemical cycles. In addition, due to their capability to assimilate atmospheric $CO_2$ into organic carbon using sunlight as the only source of energy, cyanobacteria are an interesting resource for green biotechnology. Among cyanobacteria, the strain *Synechocystis* sp. PCC 6803 is an established model organism with a broad compendium of published studies characterizing its growth and metabolism under different environmental conditions [1, 2].

Concomitant to experimental studies, computational reconstructions of metabolism play an increasingly important role to allow us to understand cyanobacterial physiology and to predict properties of metabolism and growth. Genome-scale metabolic reconstructions (GSMRs) are available for an increasing number of microbial organisms, including *Synechocystis* sp. PCC 6803 [3–10] and several other cyanobacteria [11–15].

A GSMR aims to provide a comprehensive account of the stoichiometry of the metabolic reaction network within a microbial organism. The construction of GSMRs is typically based on an available genome sequence. Annotated genes and the encoded protein complexes are linked to suitable reaction databases, such as KEGG [16, 17] or MetaCyc [18, 19] to establish gene-protein-reaction relationships. A GSMR includes all known enzyme-catalyzed metabolic reactions, transport reactions, as well as non-catalyzed processes, such as diffusion or spontaneous degradation of metabolites.

Once established, a number of computational techniques are available to analyze a GSMR and to investigate its metabolic capabilities under different environmental conditions. In particular, methods based on linear programming (LP), such as flux balance analysis (FBA) [20, 21], have become a de facto standard. The success of FBA is due to its computational simplicity, as well as the fact that its application only requires knowledge of the stoichiometry of the metabolic network, a suitable objective function, and a set of constraints on uptake fluxes—and does not require extensive knowledge about enzyme-kinetic parameters or regulatory interactions. Instead, predictions using FBA are based on the assumption of evolutionary optimality. That is, FBA seeks to predict maximal growth rates and the associated metabolic fluxes based on the assumption that an organism maximizes its growth rate given constraints on nutrient uptake rates [22, 23]. While the conditions under which the assumption of evolutionary optimally holds is still subject to considerable debate, predictions are often in good agreement with available data [23–25].

Different from heterotrophic metabolism, however, the description of phototrophic metabolism gives rise to additional challenges. Light absorption, photodamage, and the unique redox metabolism associated with photosynthesis are key aspects in a computational description of phototrophic growth. Compared to heterotrophic growth, only few studies provide a quantitative computational analysis of light-limited phototrophic growth and integrate

physiologically relevant photosynthetic properties into large-scale models of light driven metabolism [26].

The purpose of this work is to provide an updated metabolic reconstruction of the cyanobacterium *Synechocystis* sp. PCC 6803 and its quantitative analysis using previously published growth experiments [1]. Specifically, we seek to describe light-limited growth of the cyanobacterium *Synechocystis* sp. PCC 6803 and the associated energy and redox balances. To this end, we introduce a novel approach to describe light absorption within the framework of FBA that also accounts for the effects of photoinhibition. We show that the resulting description is capable of predicting quantitative properties of phototrophic growth, in particular photosynthetic oxygen release, as well as the ratio between linear and cyclic electron transport. The results demonstrate that current genome-scale metabolic models, together with appropriate constraints, are suitable to describe and predict quantitative aspects of phototrophic growth. Our method is based on only few additional parameters that have a clear biological interpretation in the context of phototrophic growth and whose numerical values can be determined from a measured growth-irradiance curve.

## Results and discussion

### Network reconstruction and FBA

The metabolic model of the cyanobacterium *Synechocystis* sp. PCC 6803 is based on previously published reconstructions, in particular Knoop et al. (2013) and Knoop et al. (2015) [5, 27]. The current reconstruction includes revised gene-protein-reaction associations, revised stoichiometric balances, and an increased coverage of metabolic processes. Additional reactions include the Entner-Doudoroff pathway [28], enabling the oxidation of glucose to gluconate and its subsequent phosphorylation to 6-phosphogluconate, as well as additional reactions for biotin metabolism and purine and pyrimidine metabolism. Additions are based on primary literature, previous reconstructions [7, 10], as well as on a comparison with the KEGG database (reference genome BA000022) [29]. A list of added reactions and their associated genes, compared to Knoop et al. (2015) [27], is provided as S2 Table.

The reconstruction consists of a total of 920 reactions, 809 metabolites (707 unique metabolites), and covers 783 genes. The 920 reactions include 865 mass balanced intracellular reactions, as well as the supply of nutrients into the growth medium (denoted as extracellular space in Fig 1) and biochemical interconversion therein, such as $CO_2$ to $HCO_3^-$. Metabolites and reactions are organized into 7 compartments: extracellular space/growth medium (e), periplasm (p), cytoplasmic membrane (cm), cytoplasm (c), carboxysome (cx), thylakoid membrane (um), and thylakoid lumen (u). For details, see Fig 1. The model was tested for stoichiometric consistency using MEMOTE [30]. Details of the network reconstruction and analysis are provided in the Materials and Methods. An annotated SBML file is available as S1 File.

Analysis of a GSMR using FBA typically requires defining a biomass objective function (BOF). The BOF specifies the amounts of cellular components needed for the synthesis of one gram cellular dry mass (gCDM). In the following, we use the (static) BOF defined in previous reconstructions as a reference [3, 5], see Table A in S1 Text, and compare the results with an experimentally obtained light-dependent BOF. The latter is derived from measurements of the mass fractions of protein and glycogen as a function of light intensity, the remaining components are scaled accordingly.

All analysis is based on a coherent set of growth experiments, reported previously [1] and summarized in the Materials and Methods, Section "Experimental data used in the analysis". Measured quantities include the specific growth rate as a function of light intensity, the

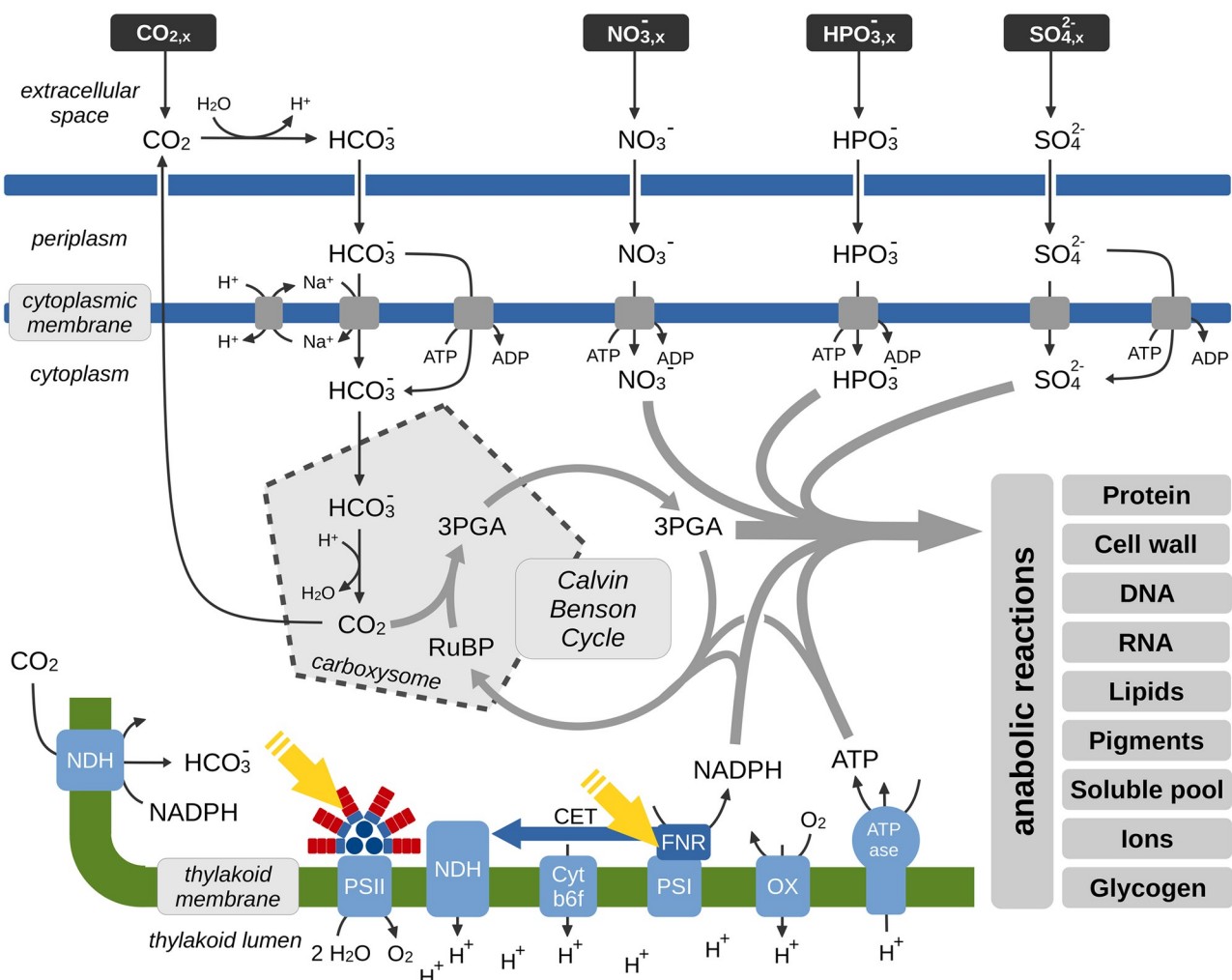

**Fig 1. Key metabolic processes involved in cyanobacterial phototrophic growth.** Growth of *Synechocystis* sp. PCC 6803 is characterized by the net uptake of bicarbonate $HCO_3^-$ from the extracellular space. Within the carboxysomes, $HCO_3^-$ is converted into $CO_2$ by the carbonic anhydrase, $CO_2$ is then assimilated by the ribulose-1,5-bisphosphate-carboxylase/-oxygenase (RuBisCO). Organic carbon is utilized to synthesize cellular biomass using *ATP* and *NADPH* regenerated by the photosynthetic electron transport chain. Cellular biomass consists of protein, cell wall components, DNA, RNA, lipids, pigments, a soluble pool (metabolites and co-factors), ions, and glycogen as a storage component. The reconstruction contains 865 mass-balanced intracellular reactions and covers 783 genes. The reconstruction consists of 7 compartments: extracellular space/growth medium, periplasm, cytoplasmic membrane, cytoplasm, carboxysome, thylakoid membrane, and thylakoid lumen. Respiratory complexes in the cytoplasmic membrane are not shown. Abbreviations: ribulose-1,5-bisphosphate (RuBP), 3-phosphoglycerate (3PGA), NADH dehydrogenase-type complex (NDH), cytochrome b6f complex (Cyt b6f), photosystem I (PSI), photosystem II (PSII), terminal oxidase (OX), ferredoxin-NADP+ reductase (FNR), cyclic electron transport (CET).

respective changes in biomass composition, light-dependent oxygen ($O_2$) evolution, as well as $O_2$ consumption in darkness. Fig 2 summarizes the data.

## Network properties and overall stoichiometry

Using the light-dependent BOF, the metabolic reconstruction of *Synechocystis* sp. PCC 6803 allows for phototrophic growth with atmospheric $CO_2$ as sole source of carbon (taken up as bicarbonate $HCO_3^-$ from the extracellular medium). When maximizing the BOF with nitrate ($NO_3^-$) as the sole source of nitrogen, the metabolic reconstruction gives rise to the following

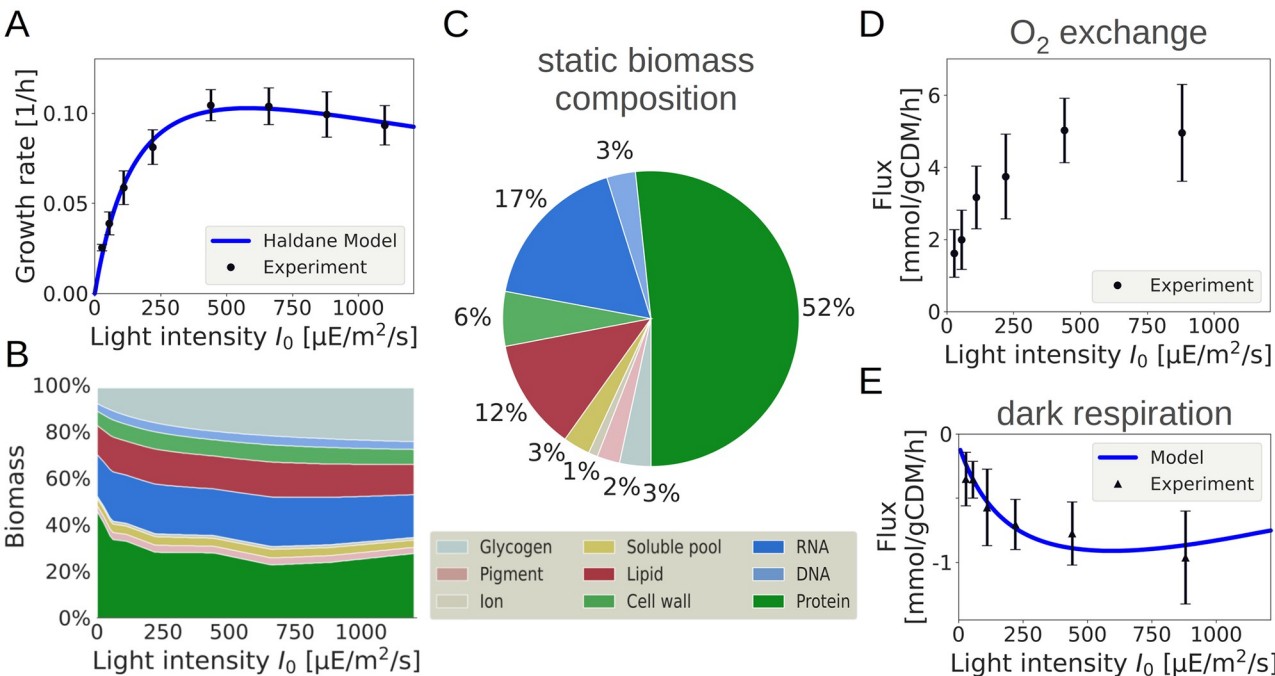

**Fig 2. Summary of the growth properties of *Synechocystis* sp. PCC 6803 obtained in a light-limited turbidostat [1].** A: The specific growth rate as a function of light intensity. The functional form of the growth rate is consistent with a phenomenological Haldane/Aiba equation (solid line). B: The light-dependent biomass composition (BOF), derived from the measured protein and glycogen mass fractions reported in Zavřel et al. [1]. C: A static biomass composition as a reference, as used in previous analyses [5]. D: Measured net oxygen ($O_2$) evolution as a function of the light intensity. E: Measured oxygen ($O_2$) consumption as a function of light intensity measured shortly after stopping illumination. The measured $O_2$ consumption in darkness is used as a proxy for respiration during illumination. The solid line shows the $O_2$ consumption of the fitted model. Growth data were originally described in Zavřel et al. [1], the experimental setup is summarized in the Materials and Methods. In the following, we use μE as an abbreviation for μmol photons in the units of light intensity.

overall stoichiometry for the synthesis of one gram cellular dry mass (see also Table B in S1 Text for the elemental composition),

$$511.73 \text{ mmol photons} + 40.85 \text{ mmol CO}_2 + 6.48 \text{ mmol NO}_3^- + 0.89 \text{ mmol HPO}_3^{2-} +$$
$$123.80 \text{ μmol SO}_4^{2-} + 221.80 \text{ μmol K}^+ + 36.18 \text{ μmol Mg}^{2+} + 8.88 \text{ μmol Fe}^{3+} +$$
$$9.79 \text{ μmol Fe}^{2+} + 4.93 \text{ μmol Na}^+ + 5.91 \text{ μmol Ca}^{2+} + 4.25 \text{ μmol Co}^{2+} +$$
$$3.94 \text{ μmol Mn}^{2+} + 3.94 \text{ μmol Zn}^{2+} + 3.94 \text{ μmol MoO}_4 + 3.94 \text{ μmol Cu}^{2+} +$$
$$7.52 \text{ mmol H}^+ + 27.93 \text{ mmol H}_2\text{O} \rightarrow 1.0 \text{ gCDM} + 57.04 \text{ mmol O}_2 + \text{byproducts}.$$

The overall stoichiometry is estimated using the BOF for a light intensity $I_0 = 660$ μE/m$^2$/s corresponding to a growth rate of 0.10 h$^{-1}$ with a mass fraction of 23% protein and 20% glycogen (in the following, μE is used to abbreviate the unit μmol photons). The overall stoichiometry indicates that the (maximal) biomass yield necessitates the excretion of three carbon-containing byproducts: 5-Deoxy-D-ribose, dialurate, and carbon monoxide (see also Table C in S1 Text). The existence of such obligatory byproducts points to either errors or inaccuracies in the metabolic reconstruction, or the existence of biologically inevitable side products that are not recycled within metabolism. These byproducts, however, constitute less than 0.07‰ of gCDM, and are therefore unlikely to be linked to exudation of compounds observed for cyanobacterial cells [31].

According to the overall stoichiometry of growth, carbon constitutes approximately 49% of the cellular (dry) mass. The C:N content is approximately 100:16, slightly lower than the Redfield ratio of approximately 106:16, but slightly larger than the experimentally determined ratio, see also Fig D in S1 Text. The N:P ratio of approximately 7:1 is significantly below the Redfield ratio of 16:1. It is noted that the elemental composition of cyanobacteria can be highly variable. The photosynthetic quotient $PQ$, the ratio of $O_2$ evolution relative to $CO_2$ assimilation is $PQ \approx 1.4$, in good agreement with previous experimental analyses [1, 5] and estimates based on the overall stoichiometry of photosynthetic growth [32, 33].

The stoichiometric overall equation only considers the stoichiometric requirements for the synthesis of biomass and neglects cellular maintenance and other processes not related to the formation of biomass, such as photorespiration. Under these assumptions, the maximal stoichiometric yield of biomass per photon is $Y_{BM}^{max} = 1.80$ gCDM/mol photons for the static reference BOF, and ranges from 1.84 to 1.97 gCDM/mol photons for the light-dependent BOF, similar to values reported for metabolic reconstructions of other cyanobacteria [34, 35]. The differences are primarily due to the varying mass fraction of protein and glycogen in the light-dependent BOF. Stoichiometrically, a minimum of approximately 9 photons are required to produce one molecule $O_2$ during growth, slightly lower than the experimentally determined value of 11 photons per $O_2$ produced [36].

## Modeling light absorption and utilization in FBA

Our aim is to provide a quantitative description of light-limited phototrophic growth. In particular, we seek to reproduce the measured specific growth rates of the cyanobacterium *Synechocystis* sp. PCC 6803 over a wide range of light intensities, as obtained through an experimental evaluation using a highly controlled and reproducible cultivation setup [1].

As shown in Fig 2, and characteristic for phototrophic microorganisms [35, 37, 38], the light-limited growth rate as a function of the light intensity $I_0$ (measured in mol photons per area per time) can be described by the phenomenological Haldane or Aiba equation [35, 39],

$$\mu(I_0) = \frac{\mu^* \cdot I_0}{K_A + I_0 + \gamma \cdot \dfrac{I_0^2}{K_A}} \cdot \tag{1}$$

The Haldane/Aiba equation is specified by three empirical parameters, $\mu^*$, $K_A$, and $\gamma$. The parameter $\gamma$ is a dimensionless number that quantifies the inhibition of growth with increasing light intensity (photoinhibition). In the absence of photoinhibition ($\gamma = 0$), the equation is identical to a Monod equation with a half-saturation constant $K_A$ and a maximal growth rate $\mu^*$.

In contrast, within constraint-based models of phototrophic growth, the uptake and utilization of photons is typically described analogous to the uptake of nutrient molecules and, in the absence of further constraints, the growth rate scales linearly with the photon uptake rate $J_I$ (measured in mol photons per gCDM per time). In the following, we therefore propose a novel approach to incorporate light uptake and utilization into constraint-based models of phototrophic growth—with the aim to better capture the growth properties of cyanobacteria and other phototrophic microorganisms. Our approach is motivated by mechanistic models of phototrophic growth [35]. Specifically, we consider a two step process in which photons are first absorbed by the cell and are then utilized with a light-dependent *photosynthetic efficiency* (also known as light-dependent *quantum yield*).

Following the experimental setup of Zavřel et al. [1], we first consider a light source with a single wavelength (monochromatic light). The rate of photon flux is described by an incident light intensity or photon flux density $I_0$ (mol photons per m$^2$ per second) at the surface of the

culture vessel. Photons are absorbed by the culture with a proportionality factor $\alpha$ (measured in $m^2$ per gCDM). The factor $\alpha$ describes an effective area of light absorption per gCDM. The rate $J_I$ of photon uptake or photon absorption is

$$J_I = \alpha\, I_0 \;. \tag{2}$$

Photons are then utilized with a (dimensionless) quantum yield $\eta(J_I)$. The quantum yield describes the ratio of the rate of utilized photons $J_I^*$ relative to the rate of absorbed photons $J_I$,

$$\eta(J_I) := \frac{J_I^*}{J_I} \in [0, 1) \;. \tag{3}$$

Following mechanistic models of light utilization [35], we postulate that the quantum yield is a decreasing function of the rate of photon uptake,

$$\eta(J_I) = \frac{K_L}{K_L + J_I} \;, \tag{4}$$

with $K_L$ as a free parameter. A derivation of Eq (4) based on a 2-state model of photosynthesis is provided in the Materials and Methods. Analysis of the constraint-based model is then based on an effective rate of light uptake $J_I^* = \eta(J_I) \cdot J_I$ (measured in mol photons per gCDM and time),

$$J_I^* = \frac{K_L \cdot J_I}{K_L + J_I} \;. \tag{5}$$

The effective rate of light uptake $J_I^*$ serves as a constraint for the maximization of the BOF in FBA, and depends on the parameters $\alpha$ and $K_L$, as well as on the incident light intensity $I_0$. The parameter $K_L$ can be interpreted as a maximal capacity of light utilization, i.e., the maximal number of photons the cell can utilize per gCDM and time. Specifically, Eq (4) implies that under low light conditions, almost all absorbed photons are utilized, $J_I^* \approx J_I$ for $J_I \ll K_L$, whereas for high light intensities there is an upper limit to the rate of light utilization, $J_I^* = K_L$ for $J_I \to \infty$. Hence, together with the maximal biomass yield $Y_{\mathrm{BM}}^{\max}$, the parameter $K_L$ also provides an upper bound for the maximal growth rate in the absence of photoinhibition or other detrimental factors.

Both parameters, $\alpha$ and $K_L$, may depend on the specific strain and culture conditions and can be estimated from the experimental growth-irradiance curve shown in Fig 2A.

## Describing photoinhibition

To account for photoinhibition, we include a description of light-induced photodamage, in particular of the D1 protein of photosystem II. Since, within our approach, protein turnover is not explicitly modeled, we describe photoinhibition by a light-dependent ATP utilization that accounts for the increased repair and translation mechanisms in dependence of light [40]. We emphasize that photodamage is a well studied phenomenon, and there is broad experimental evidence across multiple domains of life, from cyanobacteria to eukaryotic algae and plants, that photodamage occurs at all light intensities (i.e., not only at high light intensities) and is proportional to the light intensity [41–45]. Hence, to account for photodamage, we introduce a light-dependent rate $v_D$ of ATP utilization as an additional constraint in our analysis,

$$v_D = k_d J_I \;. \tag{6}$$

The rate $v_D$ is proportional to the rate $J_I$ of light absorption, with a (dimensionless)

proportionality factor $k_d$, and accounts for the increased ATP demand as a consequence of increased protein turnover and repair.

Within the model, the description of light absorption, utilization, and photodamage therefore has three free parameters, $\alpha$, $K_L$, and $k_d$, analogous to the three empirical parameters of the Haldane/Aiba Eq (1). Below, we demonstrate that these free parameters can be determined from a fit of the constraint-based model to the measured specific growth rate over the full range of light intensities. Once the numerical values of the parameters are known, additional growth properties can be evaluated.

## Incorporating blue background illumination

Prior to a numerical analysis, we have to account for the specific experimental setup used in Zavřel et al. [1]. In addition to monochromatic red light ($\lambda_{max} \approx 633$ nm), supplied with an intensity $I_0$ between 27.5 and 1100 μE/m$^2$/s, the growth setup was supplemented with blue light ($\lambda_{max} \approx 445$ nm) with a constant intensity $I_b = 27.5$ μE/m$^2$/s. The reason for the additional background illumination was to avoid possible (regulatory) growth limitations resulting from the absence of short wavelength photons [1].

In the following, we therefore make use of a heuristic *ansatz* that accounts for the utilization of blue light, while avoiding additional complexity in the model description. Instead of Eq (2), the rate of photon absorption is described by

$$J_I = \alpha \left( I_0 + \alpha_b I_b \right) , \tag{7}$$

where the dimensionless parameter $\alpha_b$ quantifies the contribution of the additional (and constant) blue background intensity relative to the intensity $I_0$ of red light. Computational details are provided in the Materials and Methods.

## Additional physiological flux constraints

When estimating the maximal growth rate and other physiological properties, constraint-based models are typically subject to additional constraints that ensure biologically plausible solutions. That is, these additional constraints are part of the model definition and can not be derived from the simulation experiments.

Following previous studies [5, 27], we assume that the growth of *Synechocystis* sp. PCC 6803 is subject to a set of additional flux constraints summarized in Table 1. Firstly, we enforce a (small) flux through the ribulose-1,5-bisphosphate-oxygenase, i.e., the use of molecular oxygen $O_2$ by RuBisCO instead of $CO_2$ as a substrate, resulting in no net $CO_2$ fixation and the formation of phosphoglycolate as a byproduct (photorespiration). Following previous models [5], the flux is set to 3% of the total RuBisCO flux. The value is within the range of experimental estimates [46, 47], see also Knoop et al. [5] for a detailed discussion of photorespiration in the

**Table 1. Additional physiological constraints used in the simulations to ensure biologically plausible solutions.** Values are adopted from previous studies [5], except otherwise noted. All percentages are relative to $O_2$ production at PSII. Notes: (a) estimated in this study, (b) of $O_2$ production at PSII.

| Constraint | Reaction Abbrev. | Range | Value | Notes |
|---|---|---|---|---|
| RuBisCO | RBCh/RBPC | | 97/3 | - |
| NGAM | ATPM | | 1.5 mmol/gDW/h | (a) |
| Mehler-like | MEHLER_1 | 6–10% | 10.0% | (b) |
| term. oxidase | CYOOum, PR0011 | 10–20% | 9.1% | (a,b) |
| Mehler PSI | PR0032 | | 0.5% | (b) |
| Mehler PSII | PR0034 | | 0.5% | (b) |

context of FBA. Secondly, we enforce a non-zero flux through the Mehler and Mehler-like reactions (the former play a minor role in cyanobacteria). The values are listed in Table 1 and are chosen according to values used in previous reconstructions [5]. Thirdly, we enforce a non-zero flux through the terminal oxidase and a non-growth associated maintenance (NGAM) reaction, accounting for basal ATP utilization that is not associated with the synthesis of biomass. The constraints are summarized in Table 1 and are assumed to apply equally across all light intensities.

In previous analysis [5], the (lower bound of) flux through the terminal oxidase was assumed to be 10% of $O_2$ production at PSII and the NGAM reaction was assumed to be 1.3 mmol/gDW/h, approximately 10% of maximal ATP production. In the following, however, we consider both parameters as unknown and adjust the values using the data reported by Zavřel et al. [1].

Specifically, in addition to the $O_2$ evolution as a function of the light intensity, the $O_2$ consumption shortly after the onset of darkness was measured (Fig 2D) [1]. The measured $O_2$ consumption in darkness varies as a function of the previous light intensity and can serve as a proxy for $O_2$ consuming processes that are not directly related to PSII activity. That is, we assume that the observed non-light associated $O_2$ consumption is also present during illumination. Within the model, non-light associated $O_2$ consumption is defined as flux through the terminal oxidase, as well as $O_2$ that is used as a substrate in metabolic reactions. Hence, non-light associated $O_2$ consumption can be calculated as $O_2$ evolution at PSII minus net $O_2$ export into the extracellular medium minus flux through the Mehler-like reactions (the latter are assumed to stop rapidly when illumination stops). We note that the exact values of the flux through terminal oxidase and NGAM have no major impact on the results reported below.

Finally, based on the experimental characterization of the cellular constituents by Zavřel et al. [1], in particular the light-dependent changes in glycogen and protein content, we adjust the biomass composition for each light intensity. The results are compared to solutions obtained for the constant reference BOF (see Fig 2B, as well as Fig C in S1 Text).

### Estimation of growth parameters

Given the physiological constraints on the flux distribution listed in Table 1, we can now estimate the four parameters used in the description of light absorption and utilization, $\alpha$, $\alpha_b$, $K_L$, and $k_d$, as well as the values of $v_{\text{NGAM}}^{\min}$ and $v_{\text{OX}}^{\min}$, such that the maximal specific growth rates matches the experimental values shown in Fig 2A, and the non-light associated $O_2$ consumption matches the $O_2$ consumption after illumination stops (Fig 2D). Computational details are described in the Materials and Methods.

Fig 3 shows the growth rate as a function of the light intensity when the description of light absorption and utilization is iteratively refined. Starting with Eq (7) only, and using $\alpha$ and $\alpha_b$ as free parameters, the growth rate is a linear function of the (red) light intensity with a slope that matches the growth curve at low light intensities. Including the photosynthetic efficiency $\eta(J_I)$, Eq (3), with $K_L$ as an additional free parameter, results in a saturation of the growth rate as a function of the light intensity. With the addition of photodamage ($k_d$ as an additional parameter, Eq (6)), we obtain the final fit of the growth curve across the entire range of light intensities. The estimated parameters are $\alpha = 0.13 \pm 0.01$ [$m^2$/gCDM], $\alpha_b = 0.66 \pm 0.24$ [dimensionless], $K_L = 119.07 \pm 13.1$ [mmol photons/gCDM/h], and $k_d = 0.07 \pm 0.018$ [dimensionless].

The estimated parameters for the terminal oxidase and the NGAM reaction are $v_{\text{OX}}^{\min} = 9.1 \pm 9.04$ [%$O_2$] and $v_{\text{NGAM}}^{\min} = 1.45 \pm 0.48$ [mmol/gCDM/h], respectively. The estimated values are close to values used in previous reconstructions [5], but are subject to substantial error intervals. The latter is expected, as ATP utilization for maintenance (NGAM) is known to be

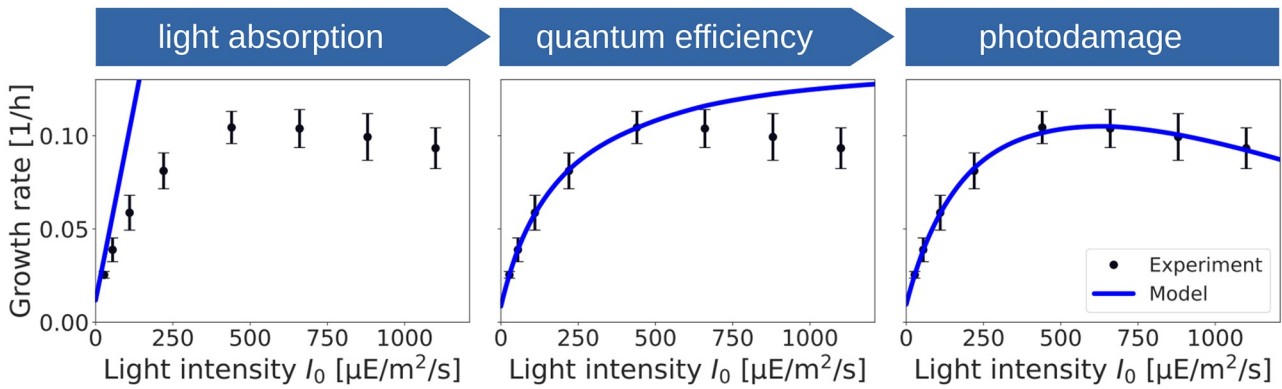

**Fig 3. Iterative refinement of the description of the specific growth rate as a function of the light intensity.** Incorporating only light absorption, Eq (7), results in a linear dependence of the growth rate on the light intensity, analogous to conventional FBA. Incorporating a variable quantum yield, Eq (3), results in saturating growth with a maximal photosynthetic capacity $K_L$. Finally, including photoinhibition, Eq (6), results in the final quantitative fit of the model to experimental values.

small in *Synechocystis* sp. PCC 6803 and the value is primarily estimated from the (offset of the) growth rate at the lowest light intensities.

We note that the detailed composition of the BOF has no major impact on the quality of the fit (see Fig A in S1 Text for a fit using the static reference BOF). The latter is consistent with previous studies showing that the detailed composition of the BOF has only a minor impact on the estimated growth rate [48]. We further note that the fitted values are specific for the strain and the experimental setup and are not necessarily universal parameters of phototrophic growth. Nonetheless, we can interpret the numerical values in the context of our analysis.

Firstly, as discussed above, the parameter $K_L$ can be interpreted as a maximal capacity for light utilization. Together with the (maximal) biomass yield $Y \approx 1.84$ to $1.97$ gCDM/mol (photons) the maximal capacity $K_L$ allows us to infer an (extrapolated) theoretical maximal growth rate of $\mu^{\text{max}} = 0.23 \text{ h}^{-1}$, corresponding to a minimal division time of approximately 3.0 h. However, photoinhibition and other detrimental effects prevent that this extrapolated maximal growth rate can be realized experimentally. The extrapolated maximal growth rate is slightly lower than the minimal division time reported for *Synechocystis* sp. PCC 6803 observed under optimal conditions [49]. We consider it an open question to what extent the maximal capacity depends on the specific experimental conditions.

Secondly, the estimated value for the contribution of blue photons $\alpha_b \approx 0.66$ is consistent with previous experimental studies. Blue light is predominately absorbed at PSI and was shown to be less efficient for *Synechocystis* sp. PCC 6803 growth compared to red light [50]. For example, it has been shown that an increase of orange/red photons (633 ± 20 nm) from 110 to 220 µE/m²/s increased the specific growth rate by 29%, whereas the same addition of blue photons (445 ± 20 nm) only increased the growth rate by 14% [51]. Likewise, cultivating *Synechocystis* sp. PCC 6803 under blue light alone resulted in reduction of maximum specific growth rate by 50–75% [50], compared to growth under orange/red light [52]. An estimated value of $\alpha_b \approx 0.66$ reflects the observed lesser contribution of blue light to growth.

## Comparison with a mechanistic model of light uptake

The assumption that light uptake is directly proportional to the incident light intensity, as postulated in Eqs (2) and (7), does not correspond to a mechanistic model of light absorption.

Instead, the factor $\alpha$ integrates the contribution to light absorption from the absorption per cell, the density of the culture, and the depths of the reactor vessel into a single quantity. This approach is motivated by the fact that it can be readily applied as long as a measured dependency of the growth rate on the light intensity is available, even when detailed information about the geometry of the reactor vessel is lacking.

The use of a flat panel photobioreactor in Zavřel et al. [1], however, allows us to compare the results with a mechanistic description of light absorption that takes reactor geometry into account. Specifically, neglecting deviations due to scattering and fluorescence, light attenuation as a function of reactor depth can be approximated by the Lambert-Beer equation. In a flat-panel reactor the light intensity at depth $z$ is approximately $I(z) = I_0 \exp(-\epsilon \rho_V z)$, where $I_0$ denotes the incident light intensity at the surface, $\epsilon$ the absorption coefficient (in units $m^2/$gCDM), and $\rho_V$ the volumetric density (in units gCDM/$m^3$). Integrating over a flat panel reactor of depth $Z$ then results in a total rate $J_I^{PBR}$ of absorbed photons [34, 35, 53],

$$J_I^{PBR} = \frac{I_0 - I_Z}{\rho_A} \ , \tag{8}$$

where $I_Z$ is the light intensity at depth $Z$ (transmitted light) and $\rho_A = \rho_V \cdot Z$ is the areal biomass density (units: gCDM/$m^2$). The required quantities, the incident light intensity $I_0$, the transmitted light intensity $I_Z$, as well as the areal biomass density $\rho_A$ have been measured in Zavřel et al. [1], see Materials and methods, hence $J_I^{PBR}$ can be calculated directly from data.

Fig 4 compares the experimentally estimated photon flux $J_I^{PBR}$ to the fitted photon flux $J_I$ obtained from Eq (7). In Fig 4 the contribution of the background blue light to $J_I^{PBR}$ is assumed to range from 0 to 100% and is shown as a (small) shaded area. That is, we assume the absorption coefficient for blue light is identical to red light and the photons are added to the red light intensity with a weight from 0 to 100% (the contribution is only visible at low light intensities). The remaining variability in $J_I^{PBR}$ is primarily due to uncertainty in the estimation of the areal biomass density. Both values for the photon flux are in good agreement, implying that using a simple proportionality factor results in a similar amount of absorbed photons as the value estimated taking the reactor geometry into account. For low light intensities, the experimentally determined flux of absorbed photons slightly exceeds the amount fitted by Eq (7). For high light intensities, however, the fitted photon uptake rate $J_I$ is slightly higher than the experimentally observed absorption. To resolve the small discrepancy, we note that, among other possible reasons, such as increasing relevance of scattering and fluorescence at higher light intensities, photodamage is modeled as ATP utilization only and therefore might overestimate the ATP usage, and hence photon requirements, at high light intensities.

## Estimating the biomass yield

The estimated rate of photon absorption allows us to assess the biomass yield per photon (units: gCDM/mol photons), defined as the growth rate divided by the rate of photon absorption. Fig 5 shows the biomass yield as a function of the light intensity, and compares the values obtained for the experimental growth rates to the values estimated from the model, as well as to the stoichiometric yield obtained by conventional FBA. The latter does not take photoinhibition and light saturation into account, but still accounts for the additional physiological constraints summarized in Table 1 (hence the stoichiometric biomass yield is slightly lower than the maximal yield obtained from the stoichiometric overall equation reported above).

The experimental yield (shown as gray area) is defined as the measured growth rate divided by the experimentally determined rate $J_I^{PBR}$ of light absorption. The contribution from blue

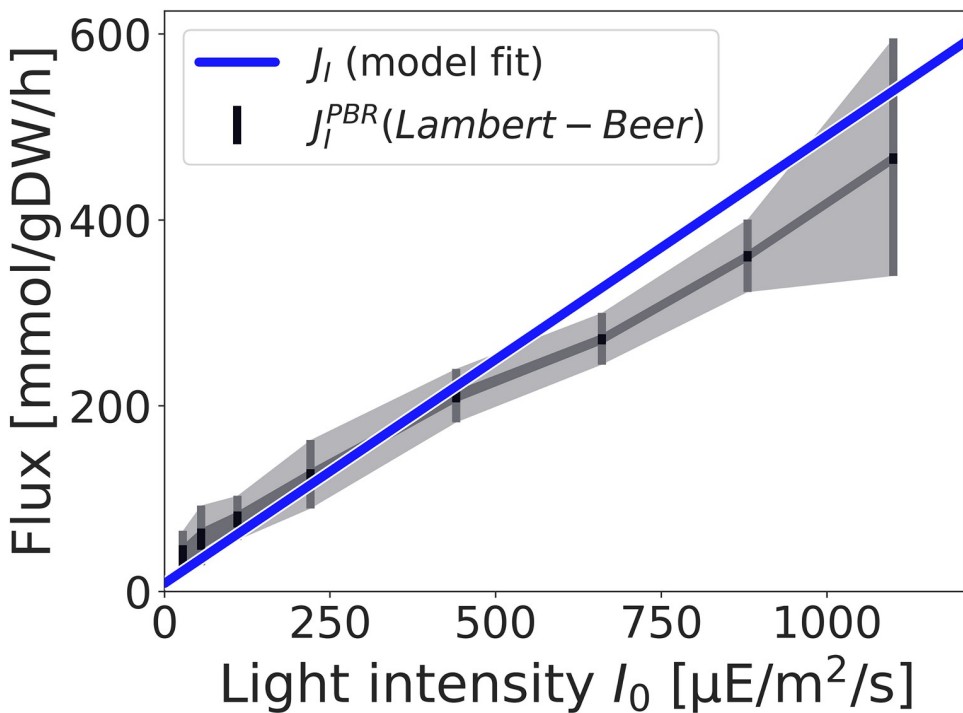

**Fig 4. Evaluating light absorption with a mechanistic model of light uptake.** Shown is the fitted photon uptake rate $J_I$, estimated according to Eq (7) in comparison with the value $J_I^{PBR}$ obtained by Eq (8). The contribution from blue photons is shown as a (small) shaded area (dark grey). The value $J_I$ is fitted using only the measured growth-irradiance curve. The value $J_I^{PBR}$ accounts for reactor geometry, transmitted light and areal biomass density. Both values are in good agreement. The light grey area in $J_I^{PBR}$ reflects uncertainty, primarily due to uncertainty in the estimation of the areal biomass density. Bars indicate experimentally measured light intensities.

light is again varied between 0 and 100% and shown as a shaded area (dark grey), while the error intervals in the estimation of $J_I^{PBR}$ are indicated in light grey. The biomass yield obtained from the model (solid blue line) corresponds to the fitted growth rate divided by the light uptake estimated according to Eq (7). The simulation takes all physiological constraints, including ATP maintenance, into account. The drop in yield for low light intensities is due to the non-growth associated maintenance requirements.

The difference between experimental and model-derived yield is primarily due to differences in the amount of light absorption at low light intensities (see also Fig 4). The estimated values are significantly below the maximal stoichiometric biomass yield, and are in good agreement with values previously reported for *Synechocystis* sp. PCC 6803. For example, Touloupakis et al. [54] report a biomass yield of $Y_{BM} \approx 1.0$ gCDM / mol photons for *Synechocystis* sp. PCC 6803 in continuous cultures using a light intensity of $I_0 \approx 150$ μE/m²/s.

It is noted that a decreasing biomass yield for higher light intensities is relevant to correctly estimate the expected phototrophic productivity in biotechnological applications [35].

## Predicting physiological properties: O₂ evolution

Using the estimated parameters, the model allows us to evaluate further physiological properties and exchange fluxes. Of particular interest is the oxygen evolution rate in dependence of the light intensity, a key property of oxygenic photosynthesis. Net O₂ evolution, as reported in Zavřel et al. [1], consists of the contribution from PSII (gross O₂ production by PSII) minus

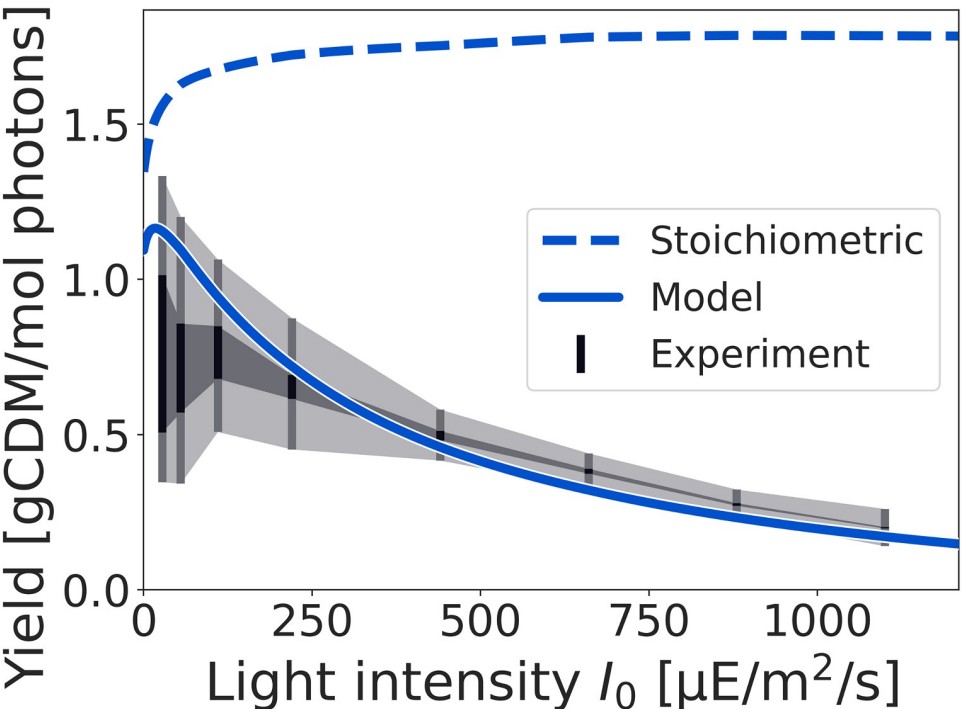

**Fig 5. Biomass yield as a function of the incident light intensity.** Shown is the stoichiometric biomass yield, as obtained by conventional FBA (dashed line). Due to non-growth associated ATP utilization, the value of the stoichiometric yield decreases for low light intensities but saturates for higher light intensities. In contrast, the yield obtained from the fitted model (blue line) decreases with increasing light intensity. The experimental yield is depicted as a shaded dark grey area that reflects the weight of blue photons from 0 to 100%. Error intervals are depicted as a light grey area. Bars indicate experimentally measured light intensities.

the use of $O_2$ for respiration, Mehler-like reactions, and the net uptake of molecular $O_2$ as a stoichiometric substrate in metabolism. We note that, as yet, the data on light-dependent net $O_2$ evolution (Fig 2D) was not used in the estimation of growth parameters, and hence can serve as a test for the consistency and the predictive value of the model.

Fig 6 shows the experimentally measured net $O_2$ evolution in comparison with the predicted net $O_2$ evolution from the parameterized model. Simulations were performed with the light-dependent BOF. We note that there is no variability in model predictions, that is, given the fitted parameters $\alpha$, $\alpha_b$, $K_L$, and $k_d$, the rate of $O_2$ evolution is fully determined. We further performed a feasibility analysis whether parameters exist that would allow us to exactly match the measured growth rate while constraining the $O_2$ evolution to measured values. Such parameters are not feasible.

As shown in Fig 6, the fitted model slightly overestimates $O_2$ production in comparison to the experimental values. The difference, while still within estimated error intervals, may stem from several factors, including potential inaccuracies in the metabolic reconstruction, as well as potential systematic errors in the experimental procedures. In particular, following the protocol described in [1, 51], experimental $O_2$ evolution was determined by stopping continuous gas supply (bubbling) and subsequently measuring accumulation of dissolved oxygen ($dO_2$) in the photobioreactor cuvette. The protocol neglects loss of $O_2$ via the headspace during the time of the measurement, and hence slightly underestimated actual $O_2$ evolution. As shown previously for the volatile product ethylene synthesized by cells containing a heterologous

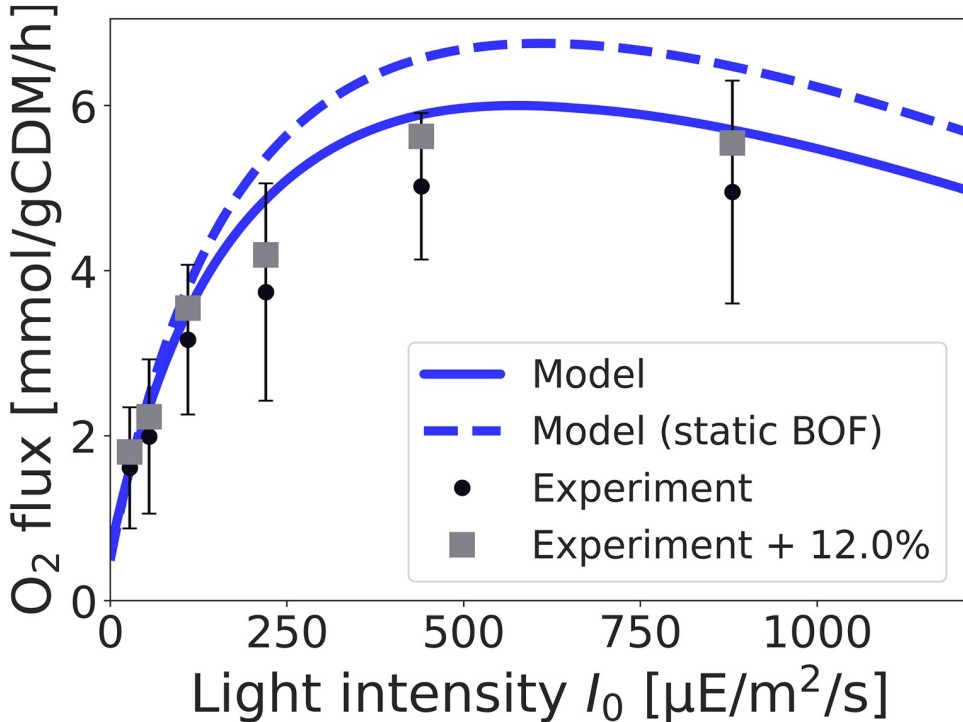

**Fig 6. Net oxygen (O₂) evolution as a function of the incident light intensity.** The prediction from the fitted model (solid blue line) is compared to experimental values (black dots). The dashed blue line shows the prediction of $O_2$ evolution obtained from the model using the static BOF. The fitted model slightly overestimates net $O_2$ production. A possible reason for the deviation is a loss of $O_2$ via the headspace of the flat-panel photobioreactor cuvette. The grey squares indicate an offset of 12% (the values are not fitted and only serve as a guide for the eye).

ethylene-forming enzyme, loss of product via the headspace can shift the measurement by up to 12% [55]. The headspace-liquid partition coefficients ($K$, the ratio of the concentration of molecules between the two phases when at equilibrium) of ethylene and oxygen are similar ($K = 0.036$ and $K = 0.03$, respectively [56, 57]), hence we can expect a similar effect for oxygen. Fig 6 illustrates the effects of a 12% systematic shift in $O_2$ evolution that would explain the observed difference (grey squares). We note that, in addition to $O_2$ loss via headspace, also other factors may play a role.

We further emphasize that using the light-dependent biomass composition (BOF) is crucial for the prediction of $O_2$ production. Fig 6 also provides a prediction of $O_2$ evolution using the static BOF (dashed line) used in previous reconstructions [5, 27], giving rise to a noticeable difference compared to the experimental values. These results support the recent analysis of Dinh et al. [48] that, while the prediction of the growth rate itself does not depend crucially on the definition of the BOF, individual reaction fluxes can be highly dependent.

## Electron transport and the ATP/NADPH ratio

In addition to $O_2$ exchange, the fitted model can be used to investigate the interplay between cyclic (CET) and linear (LET) electron transport and, more generally, the electron flow of PSII compared to PSI and the resulting ratio of synthesis of ATP relative to NADPH. Maximizing the synthesis of biomass with the given constraints gives rise to a required (optimal) ratio of

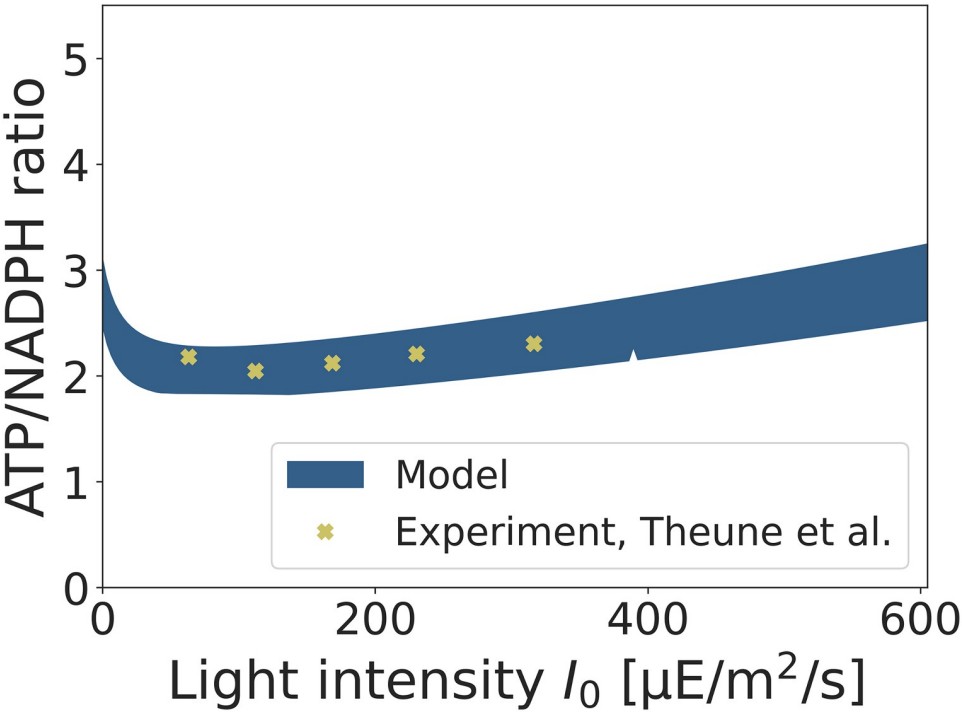

**Fig 7. The predicted range of (optimal) ATP versus NADPH synthesis estimated from the fitted model, shown as a shaded area.** The variability is due to substitution of NADPH by other redox carriers, the lower bound corresponds to the case with NADPH as the primary redox carrier. The predicted values are in good agreement with experimental values reported by Theune et al. [36].

generated ATP relative to NADPH. This optimal ratio exceeds the ratio synthesized by LET and requires a contribution from CET to meet additional ATP demands.

We note that calculating the required (optimal) ATP/NADPH ratio in the model exhibits variability since the redox carrier NADPH can, to some extent, be substituted by other carriers, such as reduced ferredoxin [58]. The use of alternative redox carriers is stoichiometrically equivalent and cannot be distinguished by stoichiometric constraint-based analysis alone. We therefore employ flux variability analysis (FVA) to evaluate the (range of) flux values of the ATPase and the ferredoxin-NADP$^+$ reductase (FNR) reaction. FVA allows us to estimate the predicted range of (optimal) ATP/NADPH synthesis rates as a function of the light intensity in the fitted model. The results are shown in Fig 7, with the variability indicated as a shaded area. The lower bound corresponds to the use of NADPH as primary redox carrier, i.e. using NADPH whenever stoichiometrically possible.

The predicted values are compared to recent experimental estimates by Theune et al. [36], who quantified CET and the resulting ATP/NADPH ratio in *Synechocystis* sp. PCC 6803. The experimental values are shown as X markers in Fig 7. We note that the experimental setup of Theune et al. [36] differs from the setup used in Zavřel et al. [1], hence the reported light intensities are not necessarily directly comparable (the impact of light intensity also depends on the light quality, volumetric biomass density, and reactor geometry). Nonetheless, within the range of experimentally used light intensities (up to $I_0 \approx 300$ µE/m$^2$/s), the predicted and experimental values are in good agreement, indicating that the fitted model provides a reasonable description of the energetics of cyanobacterial growth under these conditions. As a

potential test for further analysis, the model suggests that the ATP/NADPH ratio will increase with increasing light intensities, primarily due to the increased requirement of repair mechanisms to alleviate photodamage. A comparison of the electron fluxes (CET versus LET) compared to the data from Theune et al. [36] is provided as Fig B in S1 Text.

### Application scenarios and further developments

Having established that our approach provides a reasonable quantitative description of cyanobacterial growth, we envision several possible application scenarios. In particular, while the present analysis was based on an extensive dataset obtained under highly controlled experimental conditions, we conjecture that our approach can be applied whenever a (typically experimentally measured) growth-irradiance curve (as shown in Fig 2A) is available.

In the simplest application scenario, our approach requires three additional empirical parameters: a light absorption coefficient $\alpha$, a maximal photosynthetic capacity $K_L$, and a rate constant $k_d$ that describes photodamage. As shown above, these three parameters reflect the three free parameters of the phenomenological Haldane/Aiba equation and can be estimated by fitting the (maximal) growth rate obtained from a constraint-based model to a given growth-irradiance curve. Since in many applications, the detailed spectral properties of the light are not known or not reported, the intensity $I_0$ of the incident light is then interpreted as an empirical property of the growth condition, and the parameter $\alpha$ is estimated using Eq (2), irrespective of the spectral composition. Constraint-based optimization is then based on (an upper limit on) the rate $J_I^*$ of productively utilized photons. Within such an application scenario, additional physiological constraints, as listed in Table 1, are sourced from literature and not estimated from data.

Our approach then allows the user to infer additional growth properties that are not readily accessible by conventional FBA, such as the (decreasing) biomass yield as a function of the light intensity, or the maximal photosynthetic capacity $K_L$. It would be of particular interest to investigate to what extent the inferred parameters depend on the specific experimental conditions, and how, for example, the maximal photosynthetic capacity $K_L$ (and hence the extrapolated minimal division time) differs between experimental conditions and between cyanobacterial strains, including fast-growing cyanobacteria [59–61].

For more advanced applications, the method can be extended to account for varying spectra, and their impact on growth [9, 50, 62, 63]. To this end, the description of light absorption can be generalized to account for multiple wavelengths using wavelength-specific absorption parameters $\alpha_\lambda$, similar to Eq (7). We note, however, that such a parameterization also requires additional data, preferably independently varying light intensities for different wavelengths. A further extension would be to consider maximal capacities for the two photosystems I and II independently. To this end, the light absorption can be implemented separate for both photosystems—however again at the cost of significant additional data requirements.

### Conclusions

In this work, we present a quantitative analysis of light-limited cyanobacterial growth based on an updated genome-scale reconstruction of the strain *Synechocystis* sp. PCC 6803. While constraint-based analysis of cyanobacterial metabolism is well established, in particular in the context of computational strain design and biotechnological applications [7, 27, 62, 64–66], quantitative analyses of cyanobacterial growth using genome-scale models are still scarce. In particular, compared to the stoichiometric description of heterotrophic growth, the description of phototrophic metabolism gives rise to additional challenges due to the utilization of light as a primary source of energy. As yet, most constraint-based analyses of phototrophic

metabolism consider photons analogous to nutrient molecules, an approach that fails to capture key properties of photosynthesis, such as a variable quantum yield or photodamage.

To address this challenge, our work proposes a novel method to describe light absorption and light utilization in constraint-based models of phototrophic growth. Our approach is similar to methods that consider saturable nutrient uptake rates for heterotrophic organisms [67], but is specifically tailored to describe photosynthetic light absorption. Our method is motivated by mechanistic models of photosynthesis, and the required additional parameters have a clear interpretation in the context of phototrophic growth.

As demonstrated, our approach gives rise to a nonlinear dependency of the growth rate on the incident light intensity (Fig 3). The parameterized model allows for an analysis that goes beyond conventional FBA, and includes a (decreasing) photosynthetic efficiency and (decreasing) biomass yield for increasing light intensity (Fig 5). The latter properties are relevant in the context of biotechnological applications and reactor design [35, 45, 68], and are difficult to address using conventional FBA. Supporting previous works [48], our analysis also indicates that the cellular growth rate does not crucially depend on the detailed definition of the BOF, whereas prediction of individual fluxes can be sensitive to the assumed intracellular composition.

The model is capable to quantitatively predict several physiological properties, such as the net $O_2$ evolution (Fig 6) and the required (optimal) ATP/NADPH ratio (Fig 7), without additional assumptions or additional fitting of parameters. In particular, the ratio of ATP to NADPH synthesis is a key property for biotechnological applications. Most heterologous products have an ATP/NADPH requirement that is lower than the ratio required for the synthesis of cellular biomass [27, 64].

The key conclusions of our work are twofold. Firstly, we have shown that the metabolic reconstruction, together with the available quantitative data, provide a consistent description of phototrophic growth under the given experimental conditions. That is, light absorption, $O_2$ evolution in light, $O_2$ consumption in darkness, biomass yield, ATP/NADPH production, and growth rate together give rise to a quantitative and consistent overall picture of cyanobacterial energetics, suggesting that our computational description does not miss any major fluxes or processes. The latter is not self-evident. For example, a recent model-based analysis of *Escherichia coli* aerobic growth revealed a significant mismatch between ATP produced versus ATP required, resulting in questions about possible unknown processes [69].

Secondly, our approach extends conventional FBA, while retaining most advantages of FBA in terms of computational simplicity. Our approach is applicable in all situations where a (typically experimentally measured) growth-irradiance curve is available that allows for the estimation of the three empirical parameters $\alpha$, $K_L$, and $k_d$ (see Section "Application scenarios and further developments" above). Our approach then provides additional information, such as a maximal (extrapolated) phototrophic growth rate, and gives rise to nonlinear changes in intracellular properties, such as a variable ATP/NADPH ratio (Fig 7) as a function of light intensity —changes that can not be obtained within the strictly linear framework of conventional FBA.

We consider our approach to be an intermediate method between FBA and more complex models of light absorption and phototrophic growth, e.g., genome-scale ME/RBA models [13, 23] or hybrid models that combine detailed kinetic descriptions of photosynthesis with stoichiometric models of metabolism [22]. In particular, while hybrid models that incorporate detailed mechanistic differential equations-based models of photosynthesis are a promising path towards an improved description of phototrophic growth, such models also require significant additional investment with respect to the required knowledge of parameters, and give rise to a significantly increased computational complexity. In contrast, the strength of our approach is that it keeps the computational and conceptual simplicity of conventional FBA,

while still allowing for a quantitative description of growth properties. We conjecture that, despite the expected increasing availability of RBA/ME-type models also for cyanobacterial strains [13] and the possibility of hybrid models, FBA will remain a method of choice for constraint-based analyses in the foreseeable future.

While our focus was a quantitative description of light-limited phototrophic growth based on the experimental setup described by Zavřel et al. [1], our approach can be generalized and applied also in different contexts. Applications not considered here include, for example, limitation by an environmental factor other than light intensity and the exudation of organic compounds [31].

We argue that our approach advances the quantitative analysis of light-limited phototrophic growth, and will be readily applicable in many applications that currently make use of conventional FBA. In particular, our approach demonstrates that key properties of photosynthesis, such as a variable quantum yield and photodamage, can be incorporated into established constraint-based models of phototrophic growth without sacrificing computational simplicity.

## Materials and methods

### The metabolic network of *Synechocystis* sp. PCC 6803

The metabolic reconstruction of *Synechocystis* sp. PCC 6803 is based on previous reconstructions [5, 27] and was revised according to current literature and other recent reconstructions [7, 10]. SBO annotation was added using the SBOannotator version 0.9 [70]. Compared to [27], the added reactions relate to the Entner-Doudoroff pathway [28, 71], tyrosine and phenylalanine biosynthesis via prephenate, biotin metabolism, methanol detoxification, and purine and pyrimidine metabolism. Fatty acid metabolism was revised according to the reconstruction of Joshi et al. [7]. A list of all added reactions can be found in S2 Table. We note that recent additions and changes to metabolic reconstructions of *Synechocystis* sp. PCC 6803 have no major impact on the overall results of this study.

The model is encoded in SBML format (SBML Level 3 Version 1) and is available as S1 File (SBML) and S2 File (SBML with constraints) and S1 Table (xlsx), as well as on GitHub at https://github.com/krotlkpt/syn6803. A static biomass objective function (BOF) adopted from [5, 27] and scaled to 1 gCDM, is used as a reference BOF. The stoichiometric coefficients of macromolecular components are provided in Table A in S1 Text. The MEMOTE [30] report is provided as S3 File. We note that, while all reactions, except exchange and biomass reactions, are fully mass and charge balanced, the stoichiometric consistency test fails because of compounds without assigned mass, e.g., photons.

### Flux balance analysis (FBA)

FBA utilizes linear programming, a mathematical technique for constraint-based optimization [20, 21]. FBA is based on knowledge of the stoichiometry of metabolism, as given by a stoichiometric matrix $N$, and seeks to find a (mass-balanced) vector $v$ of reaction fluxes that satisfies a set of given flux constraints while maximizing a (linear) objective function $v_{\text{BOF}}$. The canonical form of FBA is

$$\max \quad v_{\text{BOF}}$$

$$\text{s.t.} \quad N v = 0, \qquad \text{(mass balance)}$$

$$v^{\min} \leq v \leq v^{\max}, \qquad \text{(flux bounds)}$$

where the biomass objective function $v_{\text{BOF}} = c^T v$ is a linear combination of reaction fluxes, and $v^{\min}$ and $v^{\max}$ denote upper and lower bounds on the reaction rates, respectively. Flux bounds on reaction may specify maximal uptake rates of (external) nutrients, and allow us to set minimal rates of intracellular reactions that would otherwise carry zero flux, such as the rate of the RuBisCO oxygenase reaction. In our simulations, the effective rate of light uptake $J_I^*$ serves as an upper bound for the photon uptake rate, corresponding to the sum of photon utilization rates at photosystem I and II. Uptake of other nutrients is not constrained. Additional intracellular minimal fluxes are summarized in Table 1. We note that within the framework of FBA, reactions like non-growth associated maintenance (NGAM) and electron overflow through the terminal oxidases and the Mehler and Mehler-like reactions are set as (constant) flux bounds and do not emerge from the model. Hence the model, in its current form, cannot dissect their specific role as tolerance mechanisms under high light intensities.

For all simulations, COBRApy [72], version 0.29.0, with the GLPK (GNU Linear Programming Kit, version 5.0) solver was used. Scripts and further instructions are available as S4 File and on GitHub at https://github.com/krotlkpt/syn-growth-fit.

## Experimental data used in the analysis

For parameter estimation, we make use of a coherent dataset reported previously [1]. In brief, *Synechocystis* sp. PCC 6803 (sub-strain GT-L) was cultivated in a turbidostat regime (cell density $2$–$4 \times 10^7$ cells mL$^{-1}$) using a highly controlled flat-panel photobioreactor FMT-150 (Photon System Instruments, Czechia). The cultures were grown in BG-11 medium with $NO_3^-$ as a sole source of nitrogen [73], and were bubbled by air enriched with $CO_2$ (final concentration 5000 ppm). The specific growth rate was determined from an increase of optical density signal (measured at 680 nm, $OD_{680}$) as recorded by the photobioreactor, using an exponential regression model. Temperature was kept constant at 30˚C and light intensities were controlled in a range 25–1100 μE/m$^2$/s. In addition to the specific growth rate, measurements included the transmitted light $I_Z$, the areal biomass density $\rho_A$ (derived from the measured volumetric density $\rho_V$ and the depth $Z = 2.4$ cm of the reactor cuvette), $O_2$ exchange rates (measured by a Clark-type electrode), content of pigments, glycogen and protein and elemental composition. For the full list and further experimental details, see [1].

## Light utilization and variable quantum yield

We propose a novel approach to incorporate light absorption and utilization into constraint-based models of light-limited phototrophic growth. To motivate our approach, we recall that within conventional FBA, the nutrient-limited maximal growth rate can be described by the product of the uptake rate $J_I$ and the maximal biomass yield $Y_{\text{BM}}^{\max}$ (in the absence of additional constraints),

$$\mu = Y_{\text{BM}}^{\max} \cdot J_I. \tag{9}$$

The maximal biomass yield $Y_{\text{BM}}^{\max}$ depends on the stoichiometry of the metabolic reaction network.

To incorporate light absorption and a variable quantum yield, we first consider the dependence of the growth rate on the incident light intensity $I_0$ (units μE/m$^2$/s) using a phenomenological Monod equation,

$$\mu = \frac{\mu^* I_0}{K_A + I_0}, \tag{10}$$

where $\mu^*$ denotes the maximal growth rate, and $K_A$ the half-saturation constant. The Monod

equation can be (mathematically equivalent) rewritten as [35]

$$\mu = \underbrace{\frac{1}{\alpha}\frac{\mu^*}{K_A}}_{=:Y_{BM}^{max}} \cdot \underbrace{\frac{\alpha\,K_A}{\alpha\,K_A + \alpha\,I_0}}_{=:\eta} \cdot \underbrace{\alpha\,I_0}_{=:J_I}\,, \tag{11}$$

where the term $Y_{BM}^{max}$ denotes the maximal biomass yield per mol photons (see below). Using the definition $K_L := \alpha\,K_A$, we obtain

$$\mu = Y_{BM}^{max} \cdot \eta(J_I) \cdot J_I\,, \tag{12}$$

with

$$\eta(J_I) = \frac{K_L}{K_L + J_I}\,. \tag{13}$$

As noted previously [35], the light-limited growth rate of a phototrophic organism is a product of the rate of light uptake $J_I$, the (dimensionless) quantum yield or photosynthetic efficiency $\eta$, and the maximal biomass yield per photon $Y_{BM}^{max}$.

In analogy to Eq (9), we therefore make use of an effective light absorption rate $J_I^*$ that incorporates the decreasing quantum yield as a function of light absorption,

$$J_I^* = \eta(J_I) \cdot J_I = \frac{K_L}{K_L + J_I}\,J_I\,. \tag{14}$$

As outlined in the main text, the parameter $K_L$ (units: µE/gCDM/s) corresponds to a maximal capacity of light utilization. Under low light conditions, $J_I \ll K_L$, almost all absorbed photons are used productively and $J_I^* \approx J_I$, whereas for high light intensities, $J_I \to \infty$, an upper limit $J_I^* = K_L$ is reached. The latter also justifies the definition of $Y_{BM}^{max} = \mu^*/K_L$ as the maximal biomass yield.

Eq (14) is then utilized to incorporate the variable quantum yield of photosynthesis into constraint-based models of phototrophic growth. The required parameters are $K_L$ and $\alpha$. We note that our approach, similar to previous coarse-grained models [37, 38, 45], considers the quantum efficiency of the complete photosynthetic electron transport chain and does not distinguish between photosystem I and II.

### Derivation using a 2-state model of photosynthesis

Our approach can be further motivated by a 2-state model of photosynthesis. Similar to kinetic coarse-grained models of photosynthesis [37, 38, 45], we consider the activation of a photosynthetic unit (representative of the entire photosynthetic electron transport chain) by the absorption of a photon $P^0 \to P^*$ with a rate $v_1 = \sigma \cdot I_0 \cdot P^0$. The rate depends on an effective absorption area $\sigma$ per photosynthetic unit (unit: area/mol), the light intensity $I_0$, and the concentration of inactive (or open) photosynthetic units $P^0$ (unit: mol/gCDM). The relaxation $P^* \to P^0$ back to the open state provides energy for the cell and occurs with a rate $v_2 = k_2 \cdot P^*$. The growth rate can then be described by the product of $v_2$ with the biomass yield per unit energy (photon), $\mu = Y_{BM}^{max} \cdot v_2$.

Given these rate equations, the steady-state concentration of activated $P^*$ is

$$P^* = \frac{\sigma \cdot I_0 \cdot P^T}{k_2 + \sigma \cdot I_0}\,, \tag{15}$$

where $P^T = P^0 + P^*$ denotes the total concentration of photosynthetic units. Hence, using the

definitions $K_L = k_2 \cdot P^T$ (corresponding to the maximal capacity of the photosynthetic units) and $J_I = \sigma \cdot I_0 \cdot P^T$ (corresponding to the total rate of light absorption), the specific growth rate is

$$\mu = Y_{BM}^{max} \cdot \frac{K_L \cdot J_I}{K_L + J_I} \; . \tag{16}$$

Again we obtain an effective rate $J_I^*$ of light absorption,

$$J_I^* = \frac{K_L \cdot J_I}{K_L + J_I} \; . \tag{17}$$

We note that in this derivation, the empirical maximal capacity $K_L = k_2 \cdot P^T$ corresponds to the total expression of the photosynthetic electron transport chain multiplied with its rate constant (coarse-grained into a single reaction). The empirical light absorption coefficient $\alpha = \sigma P^T$ corresponds to the total expression of photosynthetic units multiplied by the absorption area $\sigma$ per photosynthetic unit.

### Parameter estimation

Estimation of parameters was performed using the nonlinear least-square algorithm from SciPy (v. 1.11.3) `Optimize` [74]. Handling of SBML files and FBA simulations were carried out with COBRApy (v. 0.29.0) [72].

In brief, the unknown parameters are $\alpha$, $\alpha_b$, $K_L$, and $k_d$, as well as $v_{NGAM}^{min}$ and $v_{OX}^{min}$. For each light intensity, the parameters $\alpha$, $\alpha_b$, $K_L$ determine the effective light uptake rate $J_I^*$. The values of $J_I^*$, as well as the parameters $k_d$, $v_{NGAM}^{min}$ and $v_{OX}^{min}$ are then part of the linear program to maximize the BOF.

The nonlinear least-square algorithm seeks to minimize the difference between the maximal growth rate obtained from the model and the experimental growth across all light intensities, and, at the same time, to minimize the difference between the rate of non-light associated $O_2$ uptake in the model compared to the experimentally determined values (Fig 2A and 2E). Within the least-square algorithm, the differences in growth rate are weighted with a factor $10^3$, and the algorithm was provided with an initial value for each parameter as well as with bounds that constrain the parameters to positive values. The results are robust with respect to the initial conditions and intervals. For each light intensity, the biomass objective function was adjusted to reflect the experimentally measured values of glycogen and protein mass fractions at the particular light intensity. The rate of light-dependent $O_2$ (Fig 2D) was not used in the fitting process. Python scripts are provided as S4 File.

### Model evaluation: $O_2$ evolution and ATP/NADPH ratio

Given the fitted parameters, the model was evaluated for the full range of light intensities. Simulations were conducted for red light intensities ranging from 0 to 1210 μE/m$^2$/s, using evenly spaced data points (in total 1000 data points). Maximizing the BOF for each light intensity resulted in estimations for metabolic fluxes and $O_2$ export, as well as $O_2$ production at PSII.

To determine the ratio of ATP/NADPH synthesis, the fluxes of the thylakoid membrane complexes ATPase and ferredoxin-NADP$^+$ reductase (FNR) were evaluated. Due to the variability of the FNR reaction, flux variability analysis (FVA) was employed to obtain the minimal and maximal values of these fluxes for each (red) light intensities ranging from 0 to 1210 μE/m$^2$/s (200 data points). Shown are values up to 600 μE/m$^2$/s. The ratio of ATP/NADPH synthesis was then calculated as $3J_{ATPase}/J_{FNR}$, due to the stoichiometry of the ATPase with 3 ATP per full cycle.

## Sensitivity analysis

The impact of the estimated parameters on the growth rate was evaluated using sensitivity analysis [20]. For all parameters, we calculated the relative difference in the maximal growth rate given a small change in the respective parameter. Formally,

$$C_k^\mu = \frac{\Delta\mu}{\mu} \Big/ \frac{\Delta k}{k} = \frac{k}{\mu}\frac{\Delta\mu}{\Delta k} \ , \tag{18}$$

where $k$ stands for the value of the respective parameter. The scaled or normalized sensitivities $C_k^\mu$ specify the relative change in the (maximal) growth rate upon a relative change in the parameter. Positive values imply the growth rate increases with an increasing value of the parameter.

Computationally, the dimensionless (scaled or normalized) relative sensitivities $C_k^\mu$ were evaluated symmetrically around each value using a 2% change in the parameter, that is, $\Delta k = k^+ - k^-$ with $k^\pm = k \pm 0.01 \cdot k$. Subsequently, the resulting change in maximal growth rate was evaluated. Results of the sensitivity analysis for the parameters $\alpha$, $\alpha_b$, $K_L$, and $k_d$, as well as for the light intensity $I_0$ are shown in Fig 8.

The changes in scaled sensitivities $C_k^\mu$ reflect the different growth regimes. For low light intensities, the light absorption parameters $\alpha$ and $\alpha_b$, as well as the light intensity $I_0$ itself, have the highest impact on growth. The impact of photodamage ($k_d$) and maximal capacity ($K_L$) is low. The impact of the parameter $\alpha_b$ rapidly drops due to the low constant contribution of

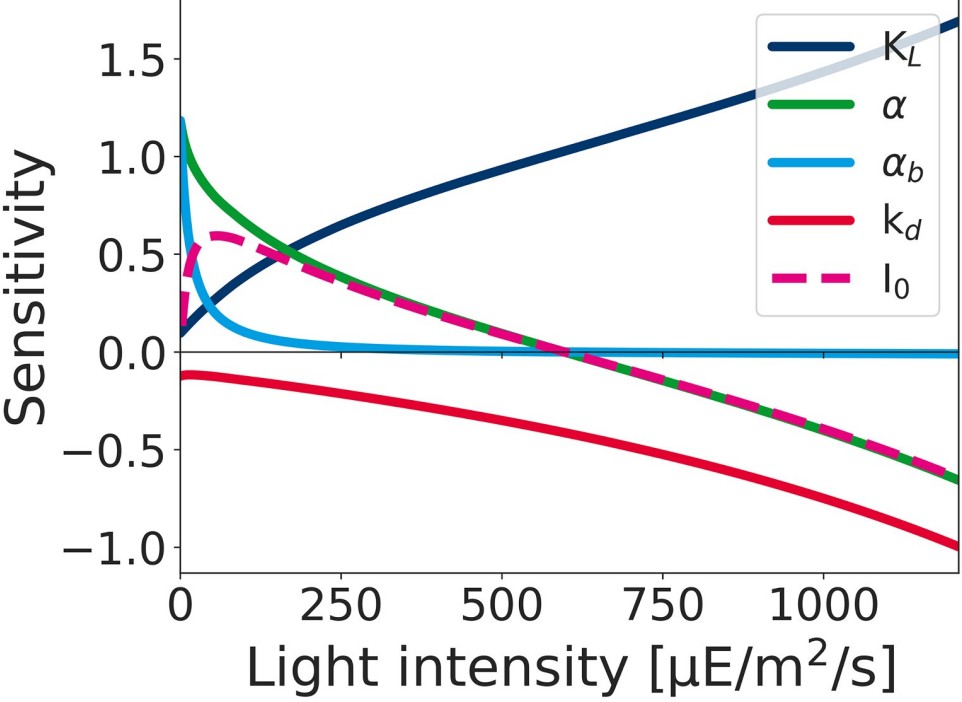

**Fig 8. Normalized sensitivity analysis of the growth rate with respect to (small) changes in estimated parameters $\alpha$, $\alpha_b$, $K_L$, and $k_d$.** The plot illustrates the different growth regimes. For low light intensities, the coefficients describing light absorption have the largest impact (light-limited growth). The sensitivity with respect to the constant blue background illumination ($\alpha_b$) rapidly drops. For high light intensities, the impact of the light absorption coefficients and the light intensity $I_0$ becomes negative. In the regime of photoinhibition, the impact of $k_d$ and the impact of the capacity $K_L$ is high.

blue light. With increasing light intensity, the impact of the maximal capacity $K_L$ and the impact of the rate constant of photodamage $k_d$ increases (the impact of $k_d$ is always negative, i.e., increasing $k_d$ will always lower the maximal growth rate). At the optimal light intensity ($I_0^{opt} = 663$ μE/m$^2$/s) the sensitivity with respect to the absorption parameter and the light intensity is zero (at this point, the contribution from blue light has no discernible impact anymore). For even higher light intensities, the cell is in the regime of photoinhibition, i.e., the impact of light intensity on the growth rate is negative and the parameter $k_d$ has a high impact.

## Supporting information

**S1 Text. Supplementary text (PDF).** Additional figures and tables as referenced in the manuscript.
(PDF)

**S1 Table. The reconstructed network (XLSX).** A table/text version of the reconstructed network in xlsx format.
(XLSX)

**S2 Table. Added reactions (CSV).** A list of added reactions compared to the model of Knoop et al. (2015) [27].
(CSV)

**S1 File. The main network reconstruction (SBML).** The genome-scale reconstruction of *Synechocystis* sp. PCC 6803 as an SBML file.
(XML)

**S2 File. Constrained network (SBML).** A model of *Synechocystis* sp. PCC 6803 as an SBML file with explicit flux constraints.
(XML)

**S3 File. The MEMOTE report (HTML).**
(HTML)

**S4 File. Script archive (TAR.GZ).** A compressed archive of scripts for simulations and to generate the figures.
(GZ)

## Author Contributions

**Conceptualization:** Rune Höper, Ralf Steuer.

**Data curation:** Rune Höper, Tomáš Zavřel.

**Formal analysis:** Rune Höper, Ralf Steuer.

**Funding acquisition:** Ralf Steuer.

**Investigation:** Rune Höper, Daria Komkova, Ralf Steuer.

**Methodology:** Ralf Steuer.

**Project administration:** Ralf Steuer.

**Resources:** Tomáš Zavřel.

**Software:** Rune Höper, Daria Komkova.

**Supervision:** Ralf Steuer.

**Visualization:** Rune Höper, Tomáš Zavřel, Ralf Steuer.

**Writing – original draft:** Rune Höper, Ralf Steuer.

**Writing – review & editing:** Rune Höper, Daria Komkova, Tomáš Zavřel, Ralf Steuer.

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
