## [Decision Letter · Decision Letter 0]

16 Feb 2024

Dear Dr Steuer,

Thank you very much for submitting your manuscript "A quantitative description of light-limited cyanobacterial growth using flux balance analysis." for consideration at PLOS Computational Biology.

As with all papers reviewed by the journal, your manuscript was reviewed by members of the editorial board and by several independent reviewers. In light of the reviews (below this email), we would like to invite the resubmission of a significantly-revised version that takes into account the reviewers' comments. Please pay particular attention to the concerns of reviewer 1 concerning consistency checks of the model.

We cannot make any decision about publication until we have seen the revised manuscript and your response to the reviewers' comments. Your revised manuscript is also likely to be sent to reviewers for further evaluation.

Sincerely,

Christoph Kaleta

Academic Editor

PLOS Computational Biology

Mark Alber

Section Editor

PLOS Computational Biology

Reviewer's Responses to Questions

**Comments to the Authors:**

Reviewer #1: In the manuscript “A quantitative description of light-limited cyanobacterial growth using flux balance analysis” submitted for publication in PLOS Comp Bio, the authors present a genome-scale metabolic model for the cyanobacterium Synechocystis sp. PCC 6803 to understand the effect of light intensity on the growth rate. The model includes several updates from previously reconstructed models from the same research group. To understand how light intensity (and composition) affects growth rate, the model was coupled to a comprehensive mathematical description of light utilization and photoinhibition dependent on the light intensity (and composition; red + blue light). Through experimental measurements, free parameters in this mathematical description were determined, as well parameters such as NGAM, so the uptake of light was constrained accordingly. The simulated growth rates, oxygen production, and ATP/NADPH ratios agreed with experimental values.

MAJOR COMMENTS:

As the authors established a phenomenological effect of light on the growth rate, it is expected that aspects of Synechocystis biology are disregarded from the model and from the subsequent discussion of the results. For instance, the relief of electron overflow through the terminal oxidases and the Mehler reactions was assumed constant for every light intensity. Readers would benefit from a discussion on how they could affect current simulation, and how further dissect their specific role as tolerance mechanisms to (high) light intensities, if they play any significant role and if they are adaptable through active gene expression.

Finally, readers would benefit from more detailed information. For instance, the experimental setup is mostly referenced and only the light intensity and composition is provided in the manuscript. Some paragraphs are vague and devoid of meaningful information, for instance, Section 2.1, paragraph 5.

Did the authors perform experiments as in Zavřel et al. or just copied data from Zavřel et al.? If the former, please include a section in Material and Methods; if the latter, please briefly describe the experimental setup and recovered data in Results, especially the medium composition.

The MEMOTE report shows that three reactions are missing a GPR. Although it is highly suspicious that such a low number of reactions to have no GPR, the reactions reported are RBPC (RuBisCO Carboxylase), BM0010 (Glycogen component of biomass reaction), RBCh (RuBisCO Oxygenase). The authors associated “Biomass” as GPR in BM00XX reactions, but RBPC and RBCh are clearly enzymatic reactions.

Following on this, the authors reported the modeling of the Entner-Doudoroff pathway, but KEGG reports no known Synechocystis gene with EC 4.2.1.12 activity. Similarly, KEGG reports no known Synechocystis BioC and BioH enzymes (biotin pathway), and there are unknown enzymes in the phenylalanine, tyrosine, tryptophan pathway.

Supplementary File S5 lacks information supporting new reactions and their gene associations. For instance, slr1229 is associated with transport reactions from cytoplasm to carboxysome (and vice versa), however, KEGG/UniProt/Cyanobase agree that slr1229 can be a sulfate permease. Similarly, why is an aquaporin (slr2057) associated with PQ transport. Moreover, why are “slr1908 or slr1841 or sll1271 or sll1272 or sll0772 or slr0042 or sll1550” associated with many, unrelated transport reactions (e.g., slr1908 is annotated as hypothetical in KEGG)?

MEMOTE reports charge balance to be correct for 59.2% of the reactions, many involving ATP hydrolysis that miss a proton to balance the reaction charge. This clearly shows that metabolite formulas are incorrect (e.g., ATP [neutral] formula is C10H16N5O13P3, but the nominal charge in the model is -4, so the [charged] formula in the model should be C10H12N5O13P3).

There is also a discrepancy as MEMOTE reports 830 reactions (not 831 as in section 2.1, second paragraph), and intriguingly, it reports only 20 mass unbalanced reactions (as the authors reports 778 are mass balanced intracellular reactions). As I can check 44 reactions are exchange reactions, how many transport reactions are in the model? Are they mass balanced as well?

Model in Supplementary File S2 has 830 reactions, while the model in S6 has 831. Please clarify.

Section 2.1, second paragraph, states stoichiometric matrix consistency was checked, but the authors did not mention the test actually failed.

Using cobrapy, the model's medium property warns that it cannot “identify an external compartment by name and choose one with the most boundary reactions”. Metabolites suffixes are not consistent with model.compartments.

Please, provide a constrained model. Supplementary File S2 simulates a growth rate of 53.24 h-1. Moreover, please provide the medium composition in the manuscript (Details are referenced to Zavřel et a.l). Why are arginine (-114.882025 mmol/gDCW/h), histidine (-4.561751 mmol/gDCW/h), “glycogen granule”, and 2-(beta-D-Glucosyl)-sn-glycerol in the model's medium? Is the nitrogen source not nitrate as it is commonly used in the BG-11 medium? Section 2.1, paragraph 4, shows a stoichiometric balance if nitrate is used as nitrogen source, but the unconstrained model shows no uptake of nitrate, but rather intake of amino acids.

Section 2.1, paragraph 5, shows vague details about the “obligatory byproducts” and the source of them (e.g., dialurate comes from the oxidative reduction of FMN). Please, add the necessary details for clarity and to show current gap knowledge in the Synechocystis’ metabolism that could be interesting to address.

Section 2.6, paragraph 2, are αb and kd unitless parameters (text shows “[n.d.]”)?

Section 2.6, paragraph 3, the NGAM estimation error opens the possibility to obtain a negative value (i.e. production of ATP). Was the parametrization constrained to avoid this situation?

Section 2.6, paragraph 5, how a theoretical doubling time of 3.0 h in the absence of photoinhibition and other detrimental effects is plausible if “photoinhibition and other detrimental effects cannot be fully alleviated”?

Section 2.7 lacks a corresponding Materials and Methods subsection. Also, there is a lack of details regarding how exactly dissolved oxygen was measured (only a reference to Zavřel et al. papers).

While the work is of great importance and interest to a broad audience, the quality and presentation of this works needs to be improved prior to publication.

MINOR COMMENTS:

Is there a reason to use “GSMR” and not GSMM or GEM, as the latter are commonly used as acronyms for genome-scale metabolic models?

Is there a reason to provide a reduced model? The reduced model depends on the protected reactions that could be different for another application of the scope of the provided reduced model.

The authors included the MEMOTE report for the whole developing branch. I locally run MEMOTE and, although the results seem to agree with Supplementary Text S6, model data should be corrected. For instance, the authors report the use of SBOannotator, however MEMOTE reports that only 14.3% of the reactions contains “SBO:0000176”. Annotation can be added manually.

Check the enumeration of the supplementary material. The MEMOTE report appears as “Supplementary Text S6” (section 4.1), but added as “S7_TEXT-MemoteReport.html” in the submission files.

Supplementary File S5 separator character (“|”) is not a common alternative when opening a csv file using spreadsheet software. Please, change the separator character to a tabulator or provide an Excel file rather than a text file.

Please, enumerate the text. Enumeration allows comprehensive comments regarding specific text in the document. I believe, as the document was prepared using LaTeX, the commands are “\\usepackage{lineno}”, then “\\linenumbers”

Supplementary Material:

Table S2 shows negative excretion flux rates, so they represent intake flux rates. Please correct the flux signs.

Why are the model provided in Supplementary Material S2 and the model provided from Supplementary Material S6 different?

Reviewer #2: In their article "A quantitative description of light-limited

cyanobacterial growth using flux balance analysis", the authors

describe a development of a genome-scale model of the cyanobacterium

Synechosystis to realistically include light harvesting, usage, and

light-induced photoinhibition. The model improvement is elegant,

because it allows defining additional constraints into the modelling

framework of constraint-based modelling, which realistically reflect

constraints of the photosynthetic machinery, and still allow treating

the computational problem with linear programming. The two main

constraints are i) a variable quntum yield, reflecting that at high

light absorbed light energy is quenched and released as heat and ii)

and light-dependent ATP requirement term, which reflects additional

demand of resources in photoinhibition due to the required protein

repair.

Overall, the paper is very clear and well written. However, it

requires a very careful spelling and grammar check, because it is full

with totally avoidable little typos, starting in the very first

sentence "The metabolism of phototrophic cyanobacterial...".

Unfortunately, I was unable to execute the Jupyter notebook. This

should urgently be improved before publication. See point 9 below.

The typos and the issues with the notebook aside, this is a very good

paper which, in my opinion, requires only minor changes. My

suggestions are as follows:

1) I would like to commend the authors for determining the biomass

composition systematically for different light intensities / growth

rates. This is rarely found and a valuable and highly interesting

experimental data set. I wonder if the authors could also give the

elemental composition of the biomass. After all, for thermodynamic

considerations, e.g. to find the energy of formation of biomass, only

the elemental composition is required. (page 6)

2) Not being a cyanobacteria expert, I am quite surprised by the

rather high PQ ratio. If 1.4 O2 are released per CO2 absorbed, where

do the additional 40% come from? The overall stoichioimetry for carbon

fixation should yield a ratio 1:1. Is the extra oxygen explainable by

the reduction of inorganic nutrients (here, nitrage, phosphate, and

sulfate) alone? (page 6)

3) Photorespiration: the authors set the flux of the RuBisCO oxygenase

reaction to 3% of the total of the RuBisCO flux, but do not give a

motivation. Where does this percentage come from? Even considering

carbon concentrating mechanisms in cyanobacteria, this seems rather

low. (page 10)

4) The experimental conditions are very specific, and while it can be

expected that fitted parameters differ under different light

conditions, possibly better reflecting natural conditions, the basic

findings are very likely of general validity. Nevertheless, it would

be nice to have a more elaborate discussion on how the model would

have to be expanded in order to realistically simulate different light

spectra. (page 12)

5) The shaded are in Fig. 5 "[...] reflects the weight of blue photons

from 0 to 100%." This is not clear to me. Earlier it was argued that a

relative efficiency of 63% fitted the data best, so why is this

necessary here? Moreover, why are there two shaded areas (light and

dark blue)?

6) "we note that photodamage is modelled as ATP utilization only, and

therefore might overestimate the ATP, and hence photon requirements,

at high light intensities" - this is very plausible. Even with a

photoinhibition rate proportional to light intensity, the additional

repair rate (and hence the additional ATP consumption) would not

increase linearly, because in higher light the proportion of already

damaged reaction centers increases, so that a smaller fraction can

receive damage. This was, for example, shown in a recent mechanistic,

differential equations-based model on photoinhibition

(https://doi.org/10.1101/2023.09.12.557336). (page 14)

7) Figure 7 shows the simulated ATP/NADPH ratio and some experimental

data points. It would also be interested, which ratio of CEF to LEF

the model predicts, because also these can, in principle, be measured

separately.

8) In the conclusions, the authors claim that their "[...] method is

motivated by mechanistic models of photosynthesis". However, as far as

I understood, this pertains only to a mechanistic model of light

absorption (section 2.7). However, there is a huge amount of

literature on detailed mechanistic models of photosynthesis, and it

would be interesting to see how the model results presented here

compare to these. That said, I do not expect the authors to perform

numerous dynamic simulations with published models, but this body o

---

## [Decision Letter · Decision Letter 1]

3 Jun 2024

Dear Dr Steuer,

Thank you very much for submitting your revised manuscript "A quantitative description of light-limited cyanobacterial growth using flux balance analysis" for consideration at PLOS Computational Biology. One reviewer noted several small points I'd like you to address in a final revision of the manuscript.

Sincerely,

Christoph Kaleta

Section Editor

PLOS Computational Biology

Mark Alber

Section Editor

PLOS Computational Biology

Reviewer's Responses to Questions

**Comments to the Authors:**

Reviewer #1: The authors have now addressed all previously raised points. Please find below comments for this revised version to consider.

MAJOR COMMENTS:

MEMOTE does not use formulas to determine if the (stoichiometric) matrix is consistent (see https://github.com/opencobra/memote/blob/develop/src/memote/support/consistency.py, lines 85-108), but I agree that photons cannot be (mass) balanced in the model and that leads to the result. In fact, Supplemental File S6, field “Minimal Inconsistent Net Stoichiometries” shows the inconsistency arises from photons imbalance, but also protons.

MINOR COMMENTS:

Please consider another format for references in Table 1. It looks like the reference “2” is the exponent (“hours square”). Maybe add another column with the references.

There are small typos, like “and and ranges from 1.84 to 1.97 gCDM/mol photons for the light-dependent BOF” in page 6, “was shown to less efficient” in page 11, “prepheneate” in page 22.

Although it is detailed that all the experimental data comes from Zavřel et al, I think that “Specifically, in addition to the O2 evolution as a function of the light intensity, the O2 consumption shortly after onset of darkness was measured” on page 9 still need to be referenced.

I think it is a stylistic choice, but I would prefer “free parameter” rather than “adjustable parameter” (as I noted a change in page 10: “with KL as an additional free parameter”). Other choices such as “GSRM” instead of “GSMM” can be settled by the editor.

In page 10, values should match the order of the parameters in the text: “The estimated parameters for the terminal oxidase and the NGAM reaction are vmin OX = 9.1 ± 9.04 [%O2] and vmin NGAM = 1.45 ± 0.48 [mmol/gCDM/h], respectively”

Another choice is “Synechocystis 6803” rather than “Synechocystis sp. PCC 6803”. The former is only used two times (page 10 and 22), while the second should be used throughout the manuscript.

On page 22, “N” is not defined. Usually, “S” is employed to denote the Stoichiometric matrix.

It might be helpful to add markers to Figure 4, showing exactly the tested light intensities. This was done for Figure S4 and others. Similar comment for Figure 5.

Is it not better to combine Figure 6 and Figure S2 to improve the comparison of model simulations? Also, the description of Figure 6 reads “an offset of ≈ 12%” while Figure S2 says “deviation of 12%” (missing the “≈” sign)

Is there a missing value in Figure 6 and Figure S2? Or oxygen exchange and dark respiration were not measured at 1100 µE/m2/s? Would these missing values affect the parameterization?

Reviewer #2: The authors addressed all comments to the reviewers to my full satisfaction.

**Have the authors made all data and (if applicable) computational code underlying the findings in their manuscript fully available?**

Reviewer #1: Yes

Reviewer #2: Yes

PLOS authors have the option to publish the peer review history of their article (what does this mean?). If published, this will include your full peer review and any attached files.

Reviewer #1: No

Reviewer #2: **Yes: **Oliver Ebenhöh

Figure Files:

Data Requirements:

Reproducibility:

References:

---

## [Decision Letter · Decision Letter 2]

26 Jun 2024

Dear Dr Steuer,

We are pleased to inform you that your manuscript 'A quantitative description of light-limited cyanobacterial growth using flux balance analysis' has been provisionally accepted for publication in PLOS Computational Biology.

Best regards,

Christoph Kaleta

Section Editor

PLOS Computational Biology

Mark Alber

Section Editor

PLOS Computational Biology

Reviewer's Responses to Questions

**Comments to the Authors:**

Reviewer #1: The authors have addressed all comments and concerns. Looking forward to see this exciting work published.

**Have the authors made all data and (if applicable) computational code underlying the findings in their manuscript fully available?**

Reviewer #1: Yes

PLOS authors have the option to publish the peer review history of their article (what does this mean?). If published, this will include your full peer review and any attached files.

Reviewer #1: No

---

## [Editor Report · Acceptance letter]

29 Jul 2024

PCOMPBIOL-D-24-00071R2 

A quantitative description of light-limited cyanobacterial growth using flux balance analysis

Dear Dr Steuer,

I am pleased to inform you that your manuscript has been formally accepted for publication in PLOS Computational Biology. Your manuscript is now with our production department and you will be notified of the publication date in due course.

With kind regards,

Anita Estes
